# Overlapping Root Architecture and Gene Expression of Nitrogen Transporters for Nitrogen Acquisition of Tomato Plants Colonized with Isolates of *Funneliformis mosseae* in Hydroponic Production

**DOI:** 10.3390/plants11091176

**Published:** 2022-04-27

**Authors:** Jingyu Feng, Weixing Lv, Jing Xu, Zhe Huang, Wenjing Rui, Xihong Lei, Xuehai Ju, Zhifang Li

**Affiliations:** 1Beijing key Laboratory of Growth and Developmental Regulation for Protected Vegetable Crops, Department of Vegetable Science, College of Horticulture, China Agricultural University (CAU), Haidian District, Yuanmingyuanxilu 2, Beijing 100193, China; mirror0406@163.com (J.F.); lvweixing@cau.edu.cn (W.L.); huangzhe@cau.edu.cn (Z.H.); ruiwenjin@163.com (W.R.); 2Beijing Agricultural Extention Station, Huixinxili 10, Changyang District, Beijing 100029, China; cnrxu@126.com (J.X.); leixihong@126.com (X.L.); 3Rural Energy and Environment Agency, Ministry of Agriculture and Rural Affairs, Beijing 100125, China; juxuehai@163.com

**Keywords:** arbuscular mycorrhizal fungi, root morphology, nutrient uptake, nitrogen transporters, nitrogen, tomato industry production

## Abstract

Understanding the impact of arbuscular mycorrhizal fungi (AMF) upon the nitrogen (N) uptake of tomato (*Lycopersicum esculentum* L.) plants is crucial for effectively utilizing these beneficial microorganisms in industrial hydroponic tomato production. Yet it remains unknown whether, besides fungal delivery, the AMF also affects N uptake via altered plant root growth or whether, together with changed N transporters expression of hosts, this impact is isolate-specific. We investigated tomato root architecture and the expression of *LeAMT1.1*, *LeAMT1.2*, and *LeNRT2.3* genes in roots inoculated with five isolates of *Funneliformis mosseae*, these collected from different geographical locations, under greenhouse conditions with nutritional solution in coconut coir production. Our results revealed that isolate-specific AMF inoculation strongly increased the root biomass, total root length, surface area, and volume. Linear relationships were found between the total root length and N accumulation in plants. Furthermore, expression levels of *LeAMT1.1*, *LeAMT1.2*, and *LeNRT2.3* were significantly up-regulated by inoculation with *F. mosseae* with isolate-specific. These results implied N uptake greater than predicted by root growth, and N transporters up-regulated by AMF symbiosis in an isolate-specific manner. Thus, an overlap in root biomass, architecture and expression of N transporters increase N acquisition in tomato plants in the symbiosis.

## 1. Introduction

Tomato (*Lycopersicum esculentum* L.) is the second most important vegetable crop next to potato worldwide. Industrial soilless cultivation of fresh market tomatoes has become popular because of the improved growth, yield, and quality of these commodities [1]. In this mode of culturing, however, the plant’s root system must develop within a small amount of growing medium; hence, the nutrient supply must be adequate and the root system must be efficient at taking up both water and nutrients [2]. Nitrogen is a critical macronutrient that plays an important role in both plant growth and metabolism. Given the high costs of N fertilizer, and the harmful impact of N fertilizer pollution on the environment, it would be desirable to develop strategies to reduce N input in agriculture production. The arbuscular mycorrhizal fungi (AMF) symbiosis can deliver substantial amounts of N to the host plant, with an estimated 21% of total N taken up by the extraradical mycelium [3]. It is well known that the N flows from the environment into the host plant via the fungus in the form of inorganic N taken up by extraradical hyphae, either as nitrate (NO_3_^−^) or ammonium (NH_4_^+^), which is then released to the plant as NH_4_^+^ [4]. Tomato is a crop known to have symbiotic partnerships with AMF, which can promote this plant’s growth and development [5]. The use of AMF in industrial tomato production may potentially strengthen the plant’s root system and their efficiency of nutrient uptake; however, in a hydroponic production system, the nutrient supply is generally less limited than that of soil conditions for vegetable cultivation. Thus, how AMF could increase plant N uptake that occurs under nutritional solution production remains unclear. One possibility is that the symbiosis changes the root architecture of host plants. Root architecture is considered as a key factor in plant productivity and its relevance to root nutrient-absorbing capacity is well-recognized [6]. Generally, AMF colonization affects root system development by increasing the extent of root branching, which in turn will increase the rate of nutrient absorption [7]. For example, when colonized by the AMF *Glomus* sp., there is greater branching of secondary roots in leek that results in an overall longer root system [8]; similarly, when *Vitis vinifera* plants are colonized by the AMF *G. fasciculatum*, the number of laterals of all orders of root increased and consequently so did the total root length [9]. In our prior study, the accrued impact of AMF on plant growth and nutrient uptake response resulted from an increasing of total lengths, surface areas, and volumes of roots [10]. Taking all into consideration, how AMF influence nutrition uptake might depend on root morphological changes elicited when colonizing their hosts.

From the plant side, roots can directly take up and use either NO_3_^−^ or NH_4_^+^ as the most important and primary source of N [11]. Over a wide range of NO_3_^−^ or NH_4_^+^ concentrations, plants may rely on several different transport mechanisms: low-affinity transport systems (LATS), which operate at high nutrient concentrations (i.e., >1 mM); and high-affinity transport systems (HATS) that predominate in the micromolar range [12,13]. Modulation of HATS and LATS functioning in coordination with modified patterns of growth and development enables plants to cope with heterogeneous N availability in soil [14]. Plant NH_4_^+^ transporters (AMTs) have been identified in *Medicago truncatula*, *Glycine max*, and *Sorghum bicolor* [15,16,17]. Root uptake of ammonium occurring at low concentrations (i.e., under high-affinity conditions) is generally mediated by *AMT1*-type ammonium transporters (AMTs) [18]. In tomato, three *AMT**1* family genes are known of, *LeAMT1.1*, *1.2*, and *1.3* [19], among which *LeAMT1.1* and *1.2* were found expressed in root hairs and leaves under N deficiency conditions, while under hydroponic growing conditions, transcript levels of *LeAMT1.2* in roots increased after NO_3_^−^ or NH_4_^+^ supplementation, whereas that of *LeAMT1.1* was inducing by N deficiency [20]. However, the effect of AMF upon the expression of ammonium transporters in tomato roots has not been reported on. Nevertheless, root-associated N_2_-fixing bacteria do excrete NH_4_^+^ to levels that can be sensed by tomato roots and this is consistent with induced expression of *LeAMT1.2* by as little as ≥1 µM external NH_4_^+^ [21]. In addition, inoculation with *Rhizophagus intraradices* induced the expression of AMTs in alfalfa root tissue [15]. Expression of the *LjAMT2.2* gene was increased more than 10-fold in the roots of *Lotus corniculatus* following inoculation with *Gigaspora margarita* [22]; that of *GmAMT1;4*, *GmAMT3;1*, *GmAMT4;1*, *GmAMT4;3*, and *GmAMT4;4* was up-regulated by *F. mosseae* [16]; likewise, that of *SbAMT3.1* and *SbAMT4* was locally induced by *F. mosseae* and *R. intraradices* in sorghum roots [17]. The AMTs are specifically expressed in arbuscule-containing cortical cells and are likely involved in the transport of mycorrhizal-derived NH_4_^+^ across the peri-arbuscular membrane into the host plant [23].

In agricultural soils, the main N source is in the form of nitrate, which is more easily mobilized and has greater fluidity in soil water flow ammonium N in soil (10–1000 times higher) [20]. The expression of four different AMF-related nitrate transporter genes in mycorrhizal *M. truncatula* and rice roots that positively affected nitrate uptake was confirmed after an increase in AMF colonization [24,25]. In tomato, *LeNRT2.3* has been functionally identified as encoding an N transporter, whose expression and assembly is induced by the presence of nitrate, to mediate low-affinity nitrate transport; it has dual roles, increasing nitrate uptake by roots and its long-distance transport from roots up to shoots; last, overexpression of this gene can increase biomass and fruit weight in tomato [26]. Furthermore, the expression of *LeNRT2.3* among the transporters assayed was higher in AMF-colonized tomato roots than in non-colonized controls [27]. Therefore, between the two N transport gene families, *LeNRT2.3* expression could be influenced via AMF–plant symbiosis, while the *LeAMT1.1* and *LeAMT1.2* effects might depend on the N status of mycorrhizal plants.

Under a nutritional solution growing condition, a deeper understanding of root architectural changes and N transporters expression patterning within this symbiosis is necessary for the commercial utilization of AMF in the tomato nutritional solution production. Furthermore, this symbiosis could offer a valuable option by which to improve nutrient uptake and enhance the N efficiency of tomato industry production.

Root growth and architecture may shift a plant’s nutritional status toward an unfavorable condition for certain genes expression; conversely, some genes expression may be favored under such less-than-favorable root growth conditions. Therefore, we hypothesized that an altered root architecture would induce the N uptake of the plant, and that this change would be associated with shifting expression levels of the *LeAMT1.1*, *LeAMT1.2*, and *LeNRT2.3* genes in the tomato–AMF symbiosis in a isolate-specific way. Taking into consideration nutrition supply and foreseeable practical aspects, here we conducted a greenhouse investigation with five isolates of *F. mosseae* from different geographical locations. These were tested under adequate nutrition supply to address three key questions: (1) How does the root growth and architecture of tomato plants change in response to this symbiosis under sufficient nutrition supply? (2) What is the capacity for increased N uptake via mycorrhizae-colonized plants, and does this depend on the isolate-specific? (3) What is the pattern of N transporters expression in root tissues and their interaction with N uptake and root growth?

## 2. Materials and Methods

### 2.1. Experimental Site and Design

#### 2.1.1. Experimental Protocol and Growing Seasons

The experiments were performed from September 2018 to March 2019, covering the autumn–winter growing season (tomato cultivar, ‘*Futesi*’ small fruit, F1, suitable for autumn-winter season and obtained from Rijk Zwaan Distribution B.V., The Netherlands), with coconut coir used as the growth medium. The experimental site was a greenhouse located at the Agricultural Technology Extension Centre in Beijing, China (40°10′ N and 116°24′ E).

#### 2.1.2. AMF Material and Inoculation

The following five AMF isolates were isolated from wild soils in different geographical locations: (1) the isolate NYN1 (109°57′ E and 39°19′ N), (2) the isolate A2 (116°10′ E and 40°08′ N), (3) the isolate T6 (129°57′ E and 46°42′ N), (4) the isolate X5 (129°46′ E and 46°54′ N), and (5) the isolate E2 (10°59′ E and 47°25′ N). They were collected using the wet-sieving method [28] and identified as *F. mosseae* [29]. The plants inoculated with the five AMF isolates were compared with the uninoculated control plants. After transplanting there was 30 g of AMF inoculant with approximately 50 spores per gram and well mixed with the growth media of coconut coir moss. The same amount of inoculant-washed microorganisms were prepared as follows: the inoculant sands had been drained with distilled water to collect possible microorganisms; then, the drainage was filtered through filter paper (Hangzhou Tezhong Zhiye GmbH, Hangzhou, China) with a particle retention potential of 5–8 mm. The washed sands were sterilized. Both the filtered solution and the residue-sterilized sand was added to the non-AMF treatments to keep the same amount of nutrition and other microorganisms from the inoculant in non-AMF treatments [28].

#### 2.1.3. Growth Conditions of Tomato Plants

Seven-week tomato seedlings were transplanted on 5 September 2018, with an average density of 2.4 plants per m^2^. The growth medium, coconut coir moss, is commercially available and is imported from Sri Lanka (by Shouguang Lvtian International Trade Co., Ltd., Shouguang City, China). This substrate is commonly used in soilless industrial tomato production in China. A split plot design with four replicates was applied with each treatment covering a surface area of 58.3 m^2^ with 140 plants. Uniform seedlings, cultivated using the same method, were randomly transplanted in the plots. The plants were grown in a solar-powered greenhouse, with day and night temperatures of 25 ± 4 °C and 16 ± 4 °C, respectively, and humidity between 60–80% under natural radiation. The plants were fertilized with a nutrient solution modified following Hoagland and Arnon (1950) that supported the growth of tomato plants [30]. The substrates (per liter) comprised 165 mg of N and 100 mg Ca added as KNO_3_ and Ca (NO_3_)_2_∙4H_2_O, 55 mg of P as KH_2_PO_4_, 220 mg of K and 65 mg S as K_2_SO_4_, 50 mg of Mg as MgSO_4_∙7H_2_O, 10.4 mg of Fe as Fe-EDTA, 10 mg of Zn as ZnSO_4_∙7H_2_O, and 10 mg of Cu as CuSO_4_∙5H_2_O based on dry substrates. The average nutritional solution supply was 1200–1600 mL per plant per day according to the plant’s growth requirements.

### 2.2. Analysis of Root Biomass, Architecture, Nitrogen Concentration, and Colonization Rates

Five plants from each subplot were randomly sampled and 20 plants were assessed per treatment on 15 January 2019 (end of the growing season). The samples of shoots were oven-dried for 48 h at 70 °C for the biomass and nutrient analyses. The roots were carefully removed from the growth medium, washed, weighed to record their fresh weight and then separated into two portions. One portion was dried to determine the biomass and nutrients, and the other soaked in 30% alcohol to analyze the AMF colonization rate. The N concentrations in the shoots were measured with the Kjeldahl digestion method. The root colonization rate was determined using the method described by Koske and Gemma [31]. The total length, surface area, and volume of roots per plant were determined with an Epson Perfection V800 Photo scanner (Epson Seiko Epson Corporation, Nagano Prefecture, Suwa, Japan), and WinRHIZO 2007 software. Yield was recorded and summed according to each harvest.

### 2.3. Root Sampling and Relative Expression of Transporters

Tomato root samples for the RNA extractions were taken randomly from each subplot, a few days before harvesting the whole plants, and these stored at −80 °C. Total RNA was isolated from these root tissues, using the ‘Quick RNA Isolation Kit’ (Huayueyang Biotechnology Co., Ltd., Beijing, China), after which their cDNA was synthesized using the ‘FastKing cDNA Dispelling RT SuperMix’ (Tiangen Biotech Co., Ltd., Beijing, China) according to the manufacturer protocol. The resulting RT reaction product was used as the template for real-time quantitative polymerase chain reaction (RT-qPCR) analysis. The RT-qPCR was run on a QuantStudioTM 6 Flex System, for which the primers were designed in Primer Premier 6 software, and all amplicons were between 80 and 200 nucleotides in size. Each DNAse treatment included three technical replicates. Owing to its consistent expression tested in tomato, elongation factors *LeUBI* and *LeEF* were used as internal reference genes (Løvdal and Lillo, 2009). The specific primer sequences were as follows:

5′-CCGCCGCTTCATACATCTGCAA (forward),5′-GCGAAACCAAGCTGCATGGAGA (reverse) for *LeAMT1.1*,5′-TTCCCTCATCTCGGCAGCTCAG (forward),5′-CCGCGTAGGTGGTGTTTGTGAG (reverse) for *LeAMT1.2*5′-GGGCTACTACACTTCCTCTGG (forward),5′- CCTCCAGCTCCTGTCATACC (reverse) for *LeNRT2.3*.5′-TCGTAAGGAGTGCCCTAATGCTGA (forward),5′- CAATCGCCTCCAGCCTTGTTGTAA (reverse) for *LeUBI* [32]5′-CTCCATTGGGTCGTTTTGCT (forward),5′-GGTCACCTTGGCACCAGTTG (reverse) for *LeEF* [33]

Each qRT-PCR reaction was carried out in a final volume of 10 µL that contained 5 µL of TB GreenTM Premix Ex TaqTM II (Takara Biomedical Technology (Beijing) Co., Ltd., Beijing, China), 0.2 µL of Rox Reference Dye II, 0.2 µL of each gene-specific primer, 1 µL of the cDNA template, and 2.4 µL of ddH_2_O. The PCR program consisted of an initial incubation at 95 °C for 30 s, followed by 40 cycles of 95 °C for 5 s and 60 °C for 30 s. The comparative threshold cycle method of ΔΔCt was adopted to quantify and analyze the relative RNA expression levels. The Ct values of the target genes were imported by the system and normalized to the Ct values of the ubiquitin by applying this equation: ΔCt = Ct target − Ct housekeeping. The fold-change was calculated from equation 2^−ΔΔCt^, where ΔΔCt = ΔCt sample − ΔCt Control [34].

### 2.4. Statistical Analysis

The data were recorded on MS Excel sheets and analyzed using the IBM SPSS software to determine mean values and standard errors. The statistical results derived from the experiment were expressed as means ± SE. The differences among the means were analyzed via a one-way ANOVA followed by Fisher least significant difference *t*-test (LSD) for multiple comparisons test (*p* ≤ 0.05) to determine if significant differences existed among the plants inoculated with AMF isolates and the uninoculated control. A univariate analysis of variance was also performed to analyze the main effects observed for the AMF isolates and the control sample. We have not compared the statistical differences of the data between two growing seasons, because of the completely different cultivars; one was big fruit and another was small fruit. The statistic *F* was used to test for the significance of the regression analysis model. The multiple coefficients of determination, R^2^, were used to test the overall effectiveness of the entire set of independent variables. In explaining the dependent variable, its interpretation is similar to that for simple linear regression: the percentage of variation in the dependent variable that is collectively explained by all of the independent variables. The relationships between root growth and N acquisition in tomato according to the data of root biomass, length, and genes represses of inoculated treatments. In these linear regressions, the *F* statistic is used for the significance of the fitted model.

## 3. Results

### 3.1. AMF Colonization and Its Effects on the Root Growth of Tomato Plants

In the present study, AMF colonization rates were between 33.8% (isolate E2) and 50.8% (isolate A2) with the nutritional solution conditions (Figure 1). Similarly, root biomass was significantly increased by at least 7.9% (isolate E2) to a maximum of 28.4% (isolate T6) when compared with uninoculated plants (Figure 2A). In terms of their root architecture, total length, surface area, and volume of roots were increased by 24.5% (isolate E2) to 119.4% (isolate A2), 29.1% (isolate E2) to 103.2% (isolate A2), and 26.0% (isolate E2) to 92.3% (isolate A2), respectively (Figure 2B–D).

### 3.2. Effects of AMF on Nitrogen Transporters Expression

As evinced in Figure 3, transcript levels of the *LeAMT1.1*, *LeAMT1.2*, and *LeNRT2.3* genes were induced in the roots of mycorrhizal plants but this differed among the isolates (Figure 3). Inoculation with isolates of E2 and T6 significantly up-regulated the expression of *LeAMT1.1*, being 1.6–1.7-fold that of the uninoculated control. Significant up-regulation of *LeAMT1.2* in response to inoculation with isolates of A2, T6, and X5 was also observed, to levels 2.6–3.0 times that of the control plants. The expression of *LeNRT2.3* was 2.0–3.8 higher in all the four isolates than the control group, the sole exception being isolate E2.

### 3.3. Influence of AMF Inoculation on Nitrogen Acquisition of Plants

The N acquisition in tomato plants was significantly correlated with their root growth, including root biomass and architecture. Given that total root length was strongly co-related to both root surface area and root volume, we only discuss the relationship between root length and N acquisition. The N concentration in root tissue was positively related to root biomass (R^2^ = 0.78, *p* = 0.02) and root length (R^2^ = 0.56) (Figure 4A1,A2); in contrast, the shoot tissue N concentration had almost no relationship with root biomass (R^2^ = 0.07), to root length (R^2^ = 0.56) (Figure 4C1,C2). The N accumulation in root tissue showed strong co-relationships with both root biomass (R^2^ = 0.91, *p* = 0.01) and root length (R^2^ = 0.80, *p* = 0.03) (Figure 4B1,B2), whereas N accumulation in shoot tissue was not so closely related to root biomass (R^2^ = 0.37) or length (R^2^ = 0.62) (Figure 4D1,D2). The N accumulation in root and shoot tissue combined was related more to root length (R^2^ = 0.63) than root biomass (R^2^ = 0.48) (Figure 4E1,E2). 

### 3.4. Nitrogen Transporters Expression Is Co-Related to Root Biomass and N Concentration in Root Tissue

As Figure 5 shows, both root biomass and N concentration were corelated to the expression levels of *LeAMT1.2* and *LeNRT2.3*, but not *LeAMT1.1*. Specifically, the root biomass (R^2^ = 0.69, *p* = 0.04) and N concentration (R^2^ = 0.62) in roots appear closely coordinated in relation to *LeAMT1.2* expression (Figure 5A1,A2), whose relationships with *LeNTR2.3* were more pronounced, with R^2^ = 0.81 (*p* = 0.01) and 0.68 (*p* = 0.04), respectively (Figure 5B1,B2); in contrast, the regressions of root biomass and N concentration to *LeAMT1.1* expression were weak and not corelated (Figure 5C1,C2).

## 4. Discussion

### 4.1. Root Growth and N Acquisition of Mycorrhizal Plants

Inoculation with isolates of *F. mosseae* resulted in the increased growth of tomato roots, especially morphological branching of their roots (Figure 2). The root biomass and architecture (i.e., lengths, surface areas, and volumes) were increased following all fungi isolates inoculation when compared with control plants, for which the plant response in terms of root architecture was more pronounced than that of the changed biomass (Figure 2). The highest total root length ensued under colonization with isolate A2, with an induction factor of 2.19, while the root biomass had an induction factor of 1.21 (Figure 2A,B). The root system characteristics altered by AMF colonization have been extensively reported in the literature, with inconsistent results found for different plant and/or fungi species. For instances, the inoculation of *Arachis hypogaea* L. or *Cajanus cajan* L. with *Gigaspora margarita* led to a more advanced development of their lateral roots, especially of second- and third-order ones, and these responses were apparent after 30 days [35]. Several AMF species—*R. intraradices*, *F. caledonium*, *Gigaspora margarita*, and *Glomus versiforme*—substantially affected the distribution of root diameter classes, increasing the proportion of fine roots (0–0.4 mm) and decreasing the proportion of coarse roots (0.6–1.2 mm) [36]. In other studies, greater root branching was recorded for *Annona cherimola* Mill. [37] and *Apium graveolens* L. [10]. The formation of AMF–plant symbiosis can alter the phenotype of the host roots, although the patterns of alteration evidently vary across host species [38]. In the presence of sufficient nutritional solution supply, even with isolate-specific impacts, the plants were not only well colonized by fungi but also incurred strong effects on their root branching. This indicated that with AMF symbiosis, tomato plant roots have a more branched root system in a nutritional solution production, albeit in an isolate-specific manner.

We found that root biomass and total root length were significantly correlated with N uptake in plants (Figure 4). In particular, the concentration and accumulation of N in root tissue is strongly correlated with the responses of root biomass and root length (Figure 4A1,A2,B1,B2). In contrast, both the relationship between N concentration and accumulation in shoot tissue were less corelated to root biomass when compared with root length (Figure 4C1,C2,D1,D2). But why root architecture is enhanced by AM inoculation remains to be elucidated. In whole plants, the N uptake appears to be more linked to changes in total root length than root biomass, although the latter is also contributing to N accumulation (Figure 4E1,E2). Increased root growth was accompanied by greater concentrations of N in both root and shoot tissues, and, after taking biomass into account, total N content in plant tissue did increase significantly (Figure 4E1). This is perhaps given that mineral nutrition is one of the key factors controlling root development, with the most striking morphogenetic power has been ascribed to N [39]. Accordingly, an increase in nutrient absorption by mycorrhizal roots may drive the root growth of plants (Figure 2A).

### 4.2. Transporter Genes LeAMT1.1, LeAMT1.2, and LeNRT2.3 Were Up-Regulated by Inoculation with AMF Isolates

The *LeAMT1.1* and *LeAMT1.2*, the two important high-affinity NH_4_^+^ transporters, were reported to have contrasting responses: *LeAMT1.1* was induced by N deficiency though *LeAMT1.2* was increased by an NH_4_^+^ or NO_3_^−^ supply [20]. While *LeAMT1.2*-mRNA levels in a concentration-dependent manner were controlled by NH_4_^+^ concentration or the plant N status, the peak expression with 2–5 µM NH_4_^+^, a further increase in NH_4_^+^ was reduced [21]. By contrast, we uncovered isolate-specific effects, in that both of these transporters’ expression still increased, albeit variously among treatments, because of the inoculation of AMF under as much as 1.8 mol L^−1^ NH_4_^+^ supply (Figure 3). The expression of *LeAMT1.1* was significantly up-regulated by inoculation with isolates E2 or T6, while the expression of *LeAMT1.2* was significantly induced by inoculation with A2, T6, and X5, in comparison with the uninoculated control (Figure 3). Other research work has reported strong inductions of *LeAMT1.1* and *LeAMT1.2* gene expression in mycorrhizal roots, evidence that host plants had NH_4_^+^ transporters that were up-regulated under AMF colonization, with the specific expression of these up-regulated AMTs genes in arbuscule-colonized cortical root cells shown in *M. truncatula* [15], *Lotus japonicus* [22], *Glycine max* [16], and *Sorghum bicolor* [17].

Similarly, in our study, the expression of *LeNRT2.3* was found significantly increased when tomato was inoculated with isolates of A2, T6, X5, and NYN1 relative to the control plants, yet the expression of this nitrate transporter depended on the isolate identity (Figure 3). It was reported that *LeNRT2.3* expression is able to respond to AMF colonization [27]; this protein functions as a low-affinity transporter, and its induced expression indicated a higher N-use efficiency in tomatoes [40]. *LeNRT2.3* was also suggested to play a key role in the xylem transport of N from root to shoot tissues and in N uptake by roots [26]. In our study, the relative expression levels of *LeNRT2.3* were correlated with both root biomass and N concentration in root tissue (Figure 5B1,B2). Taken together, the expression of *LeNRT2.3* driven by the symbiosis could be important for N-use efficiency in tomatoes. As N is a major factor determining plant growth and yield, genes likely influence plant growth by modulating N uptake rates or remobilization activity [26].

### 4.3. Overlapping Effects of AMF Upon Root Growth and N Transporters Expression

In this study, the three N transporters tested were up-regulated and those genetic responses were correlated with root biomass and N concentration (Figure 5A1,B1,B2). These results suggest that the increased N accumulation in AMF-inoculated plants is mediated by the up-regulation of their nitrate transporter genes. The up-regulation of *LeNRT2.3* is consistent with the increased shoot N uptake found in comparison with the control plants. This is similar to findings of the Saia et al. experiment, where they observed that the up-regulation of both *NRT1.1* and *NRT2* by inoculation with AMF and AMF + PGPR was consistent with the increased aboveground tissue N concentration when compared with that of non-inoculated plants [41]. The promoters of *LeAMT1.1*, *AMT1.2*, and *LeNRT2.3* were active in the root tissue of mycorrhizal plants, suggesting an increasing function for N transporters expression in the uptake of N in this symbiosis.

As descripted before, plants employ several different transport mechanisms across a wide range of nitrate concentration [42]. To date, in *Arabidopsis thaliana*, six genes have been identified that belong to the AMTs gene family; there are 10 AMTs in rice, 16 in soybean, 16 in poplar, and 3 in tomato [43]. Most nitrate uptake activity is mediated by *LeNRT1.1* and *LeNRT1.2*, but especially the latter [44]. Further studies could investigate more N transporters, not only AMF-related genes but also other genes, to identify and elucidate mechanisms responsible for N transport changing in mycorrhizae-colonized plants.

## 5. Conclusions

Our results demonstrate that inoculation with AMF can fully stimulate the biomass, length, surface area, and volume of roots in tomato plants, thereby increasing their N absorption. Root biomass and length are linearly correlated with N uptake, while the N status was closely related to the plant’s growth potential. The enhanced N content in plant tissue following AMF inoculation exceeded our expectation based on root architecture alone. In response to AMF inoculation, expression levels of the examined N transporters are up-regulated to coordinate the greater N accumulation. This study strengthens our understanding of the synergistic effects of AMF on plant root growth and nutrient uptake. Not only root growth but also an increase in root system size and branching and the promotion of N transporters expression was found, which suggests the AMF inoculation augments the N acquisition capacity of tomato plants via both root growth and gene expression. Furthermore, we know that AMF colonization is correlated with the hyphal branching network of rhizosphere volumes, enabling the mycelium to form a better distributed surface and transport N toward the host plant [45,46]. Taken together, using efficient isolates of AMF offers a promising way to improve nitrogen uptake in industrial tomato production.

## Figures and Tables

**Figure 1 plants-11-01176-f001:**
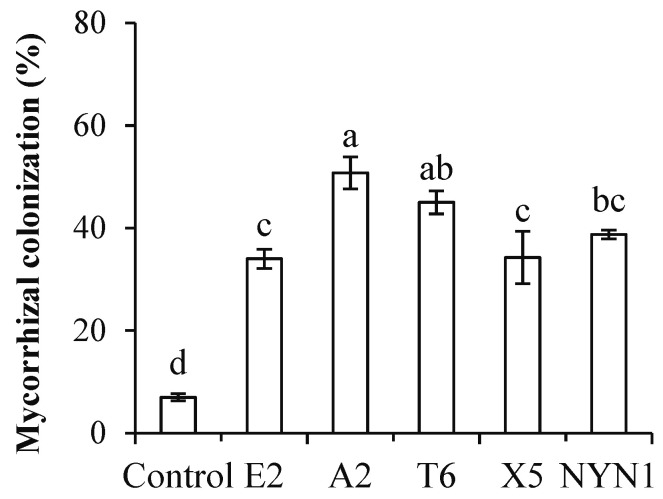
Colonization of AMF isolates. Different letters indicate significant differences (*F* = 27.58; *p* ≤ 0.05) among the different treatments after ANOVA and LSD tests. Bars are the mean ± SE (*n* = 4).

**Figure 2 plants-11-01176-f002:**
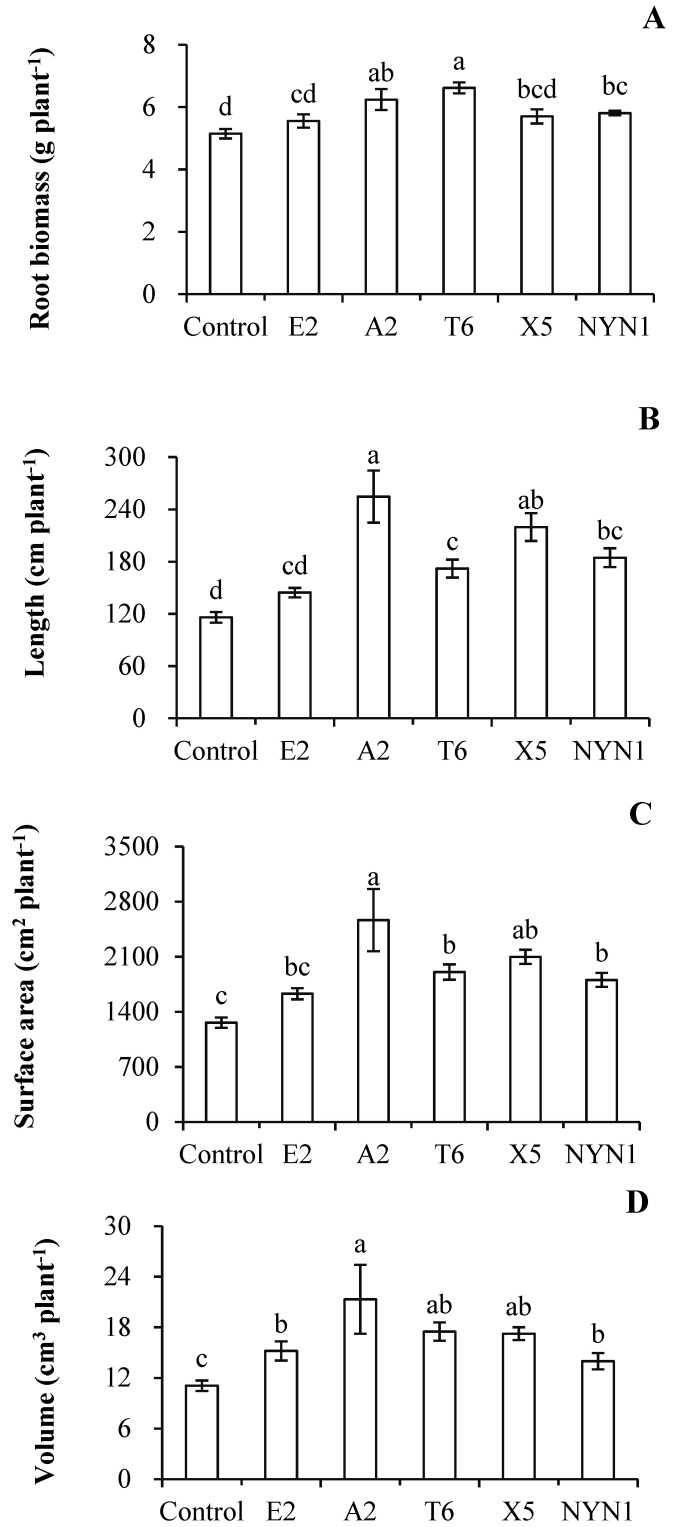
Effects of AMF isolates on root biomass and architecture of tomato plants. Root biomass (**A**), root total length (**B**), root surface area (**C**), and volume (**D**). Different letters indicate significant differences (*p* ≤ 0.05) among the different treatments after ANOVA and LSD tests. Bars are the mean ± SE (*n* = 4). *F*-_root biomass_ = 6.00; *F*-_root lengths_ = 10.52; *F*-_surface areas_ = 6.10; *F*-_volumes_ = 3.45.

**Figure 3 plants-11-01176-f003:**
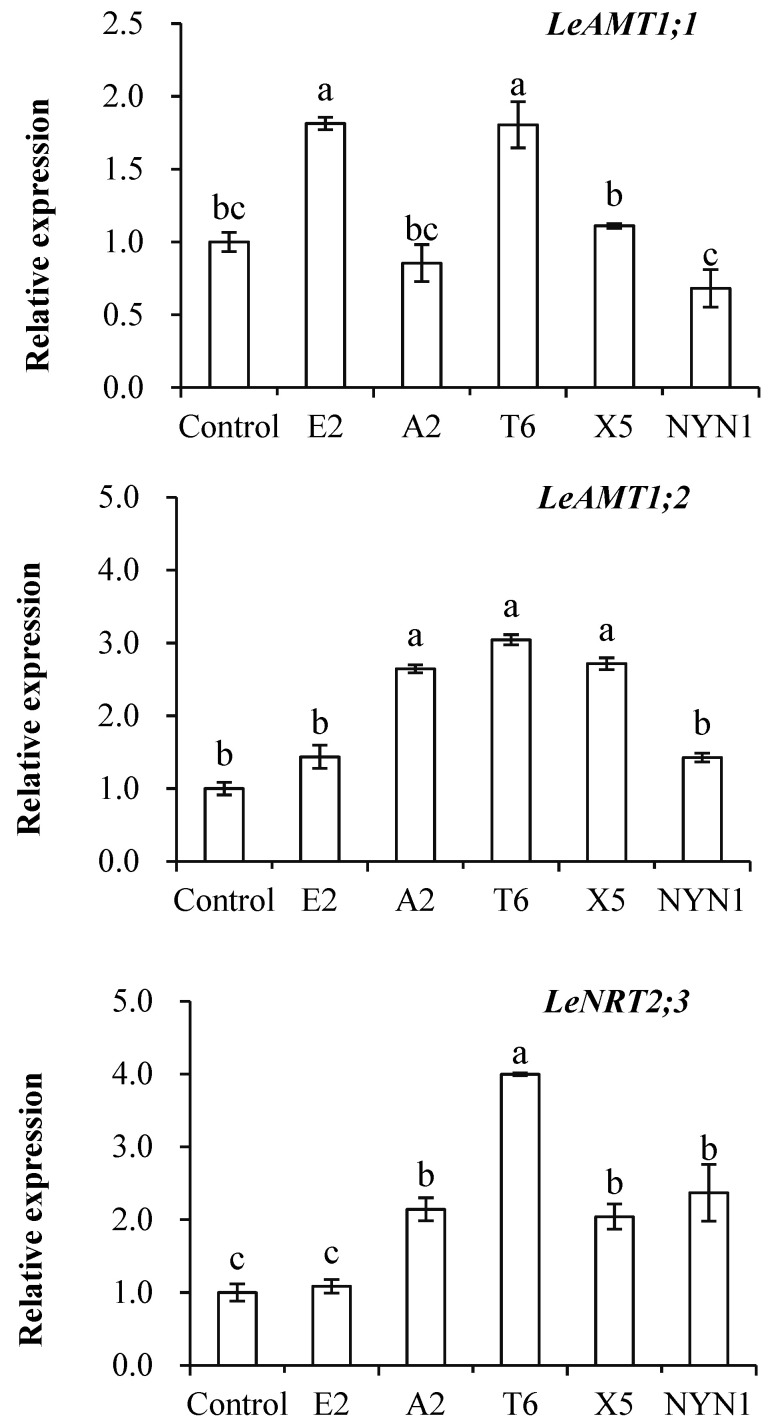
Effects of AMF on the expression of three nitrogen transporter genes. Different letters indicate significant differences (*p* ≤ 0.05) among the different treatments after ANOVA and LSD tests. Bars are the means ± SE (*n* = 4). *F*-*_LeAMT1.1_* = 16.86; *F*-*_LeAMT1.2_* = 19.85; *F*-*_LeNRT2.3_* = 19.57.

**Figure 4 plants-11-01176-f004:**
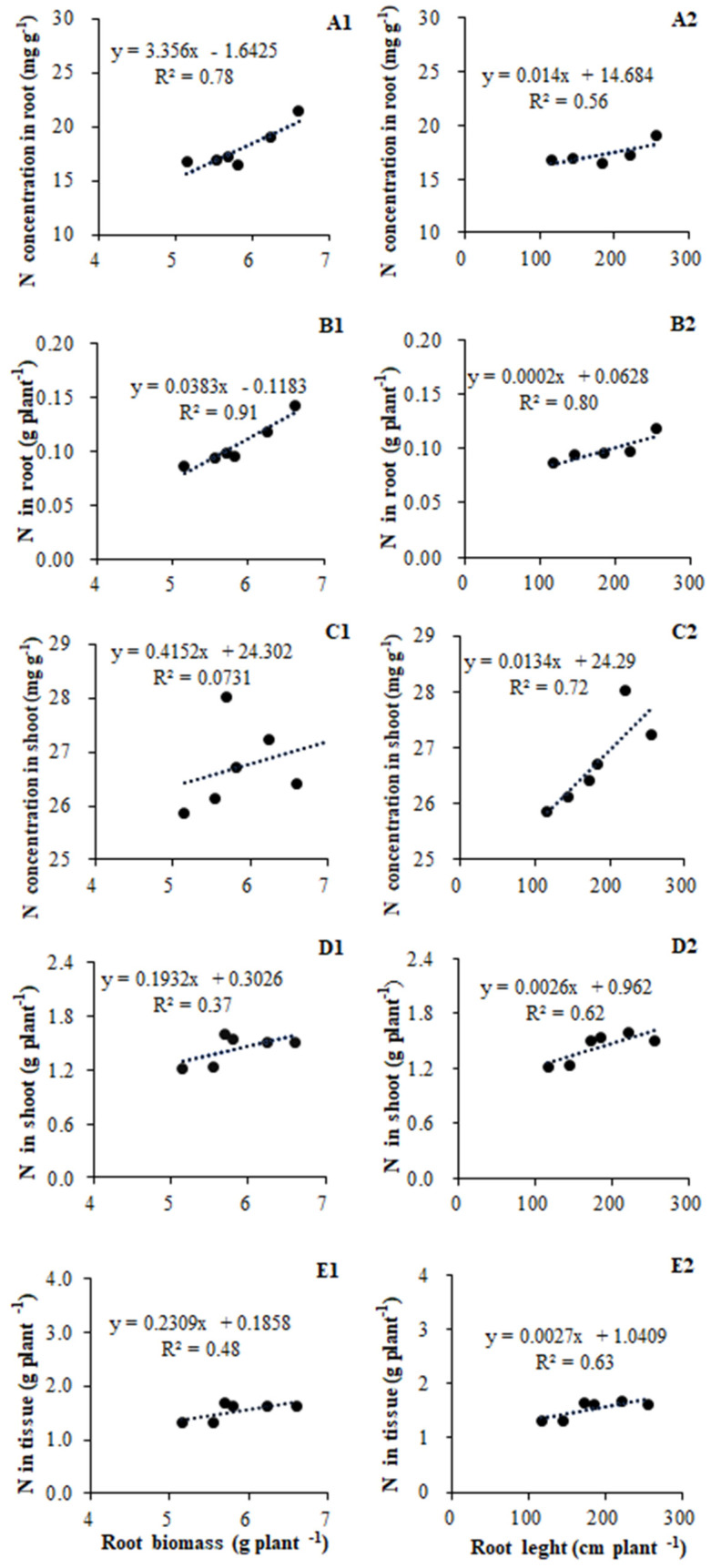
Relationships between root growth and N acquisition in tomato according to data of root biomass, length, and genes repress of inoculated treatments (**A1**,**A2**,**B1**,**B2**,**C1**,**C2**,**D1**,**D2**,**E1**,**E2**).

**Figure 5 plants-11-01176-f005:**
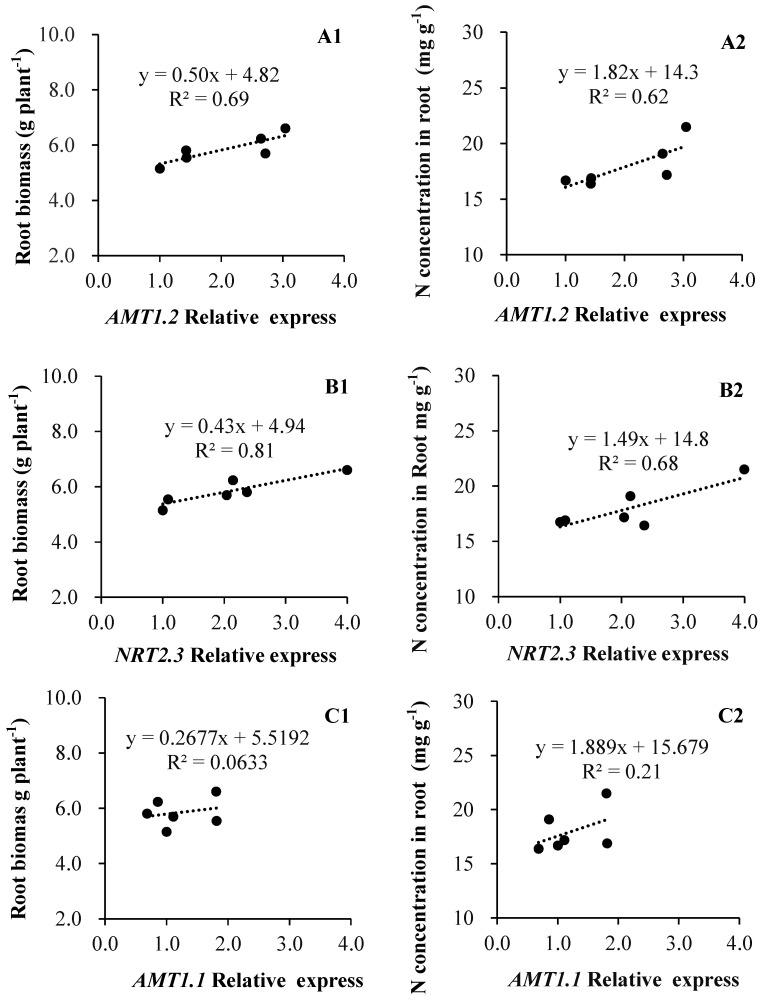
Relationships of N transporter genes expression to root biomass and N acquisition in tomato ((**A1**,**B1**,**C1**): root biomass; (**A2**,**B2**,**C2**): N concentration).

## Data Availability

Not applicable.

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
