# Peer review of "Overlapping Root Architecture and Gene Expression of Nitrogen Transporters for Nitrogen Acquisition of Tomato Plants Colonized with Isolates of Funneliformis mosseae in Hydroponic Production"

_plants, 2022, doi:10.3390/plants11091176_

Round 1
Reviewer 1 Report
Second Review on
“Overlapping root architecture and gene expression of nitrogen
transporters for nitrogen acquisition of tomato plants colonized
with isolates of Funneliformis mosseae in hydroponic production”
I am attaching my previous comments on the specific submission. In the revised form, the authors have only made some minor corrections (In two Figure Legends and a sentence in Materials and Methods), without addressing any of my comments. Therefore, I am at the unpleasant position to reject the above paper.
“This is an original article providing new information on the possible use of various fungal strains of Funneliformis mosseae in ameliorating nitrogen uptake by plants. Although the work is characterized by a certain novelty and provides important information that could be used as a starting point for providing a solution to improving tomato crop yields, there are certain points that need to be addressed and clarified in order to have it published.
My major concern is that the paper is very poorly written and the results are poorly presented. I am including the manuscript highlighted by yellow for the authors to check all the phrases that do not make sense in the manuscript (they are plenty).
There are serious flaws in the way the results are presented and interpreted as well as in the terminology used by the authors. The strains used in this study should be presented at the first paragraph of the Results and a detailed analysis of their isolation procedure as well as the methodology used for their identification should be included. In this way, it would be easier for the reader to understand Fig. 1 which should be more analytically presented and described. The authors need to clarify what exactly does this “nutrient solution” contain as far as the microorganism is concerned. When the filter was used, how is it certain that only spores of this strain are retained and not of other microorganisms too?
In Fig. 2, the authors talk about changes in root biomass and root length. Only in strain A2 there seems to be a significant increase in root biomass resulting in an increase in root volume and root surface. The authors could incorporate a photo where “root architecture” would be shown, where also root branching could be seen. In Fig.3, when the expression of the specific genes is presented the Y axis refers to “relative expression” in comparison to the control. I believe the authors should clarify in this point if this specific metric has taken into consideration the increase in the root biomass already reported. This would give an estimation of the actual increase in gene expression and not an increase which is due to the increased biomass. To conclude, I believe that in order for this work to be accepted for publication, these specific points need to be clarified as they are really important for the validity of the results presented in Fig. 3. “
Author Response
Thank very much for your comments. especially for highlight the grammar mistakes. We improved our manuscript according to you comments. Please see below:
Best regards
Zhifang LI
I am attaching my previous comments on the specific submission. In the revised form, the authors have only made some minor corrections (In two Figure Legends and a sentence in Materials and Methods), without addressing any of my comments. Therefore, I am at the unpleasant position to reject the above paper.
“This is an original article providing new information on the possible use of various fungal strains of Funneliformis mosseae in ameliorating nitrogen uptake by plants. Although the work is characterized by a certain novelty and provides important information that could be used as a starting point for providing a solution to improving tomato crop yields, there are certain points that need to be addressed and clarified in order to have it published.
My major concern is that the paper is very poorly written and the results are poorly presented. I am including the manuscript highlighted by yellow for the authors to check all the phrases that do not make sense in the manuscript (they are plenty).
We corrected. Please see:
line 45-46 to the new version at line 45-46;
line 67-76 to the new version at line 69-79;
line 91 to the new version at line 94;
line 104-108 to the new version at line 115-119;
line 114-117 to the new version at line 125-128;
line 128 to the new version at line 139;
line 133 to the new version at line 144;
line135 to the new version at line 146;
line 147-152 to the new version at line 158-164;
161-163 to the new version at line 163-224;
Line 238-240. Delected (not necessary);
Line 253-255 to the new version at line 325-327;
Line 258 to the new version at line 334;
Line 281-283 to the new version at line 359-361;
Line 358 to the new version at line 439;
Line 379 to the new version at line 461-462;
Line 425 to the new version at line 514.
There are serious flaws in the way the results are presented and interpreted as well as in the terminology used by the authors. The strains used in this study should be presented at the first paragraph of the Results and a detailed analysis of their isolation procedure as well as the methodology used for their identification should be included. In this way, it would be easier for the reader to understand Fig. 1 which should be more analytically presented and described. The authors need to clarify what exactly does this “nutrient solution” contain as far as the microorganism is concerned. When the filter was used, how is it certain that only spores of this strain are retained and not of other microorganisms too?
Thank you very much for your suggestions. We have described in our former paper, reference 29:
“The following five AM fungi isolates were collected from wild soils in different geographical locations: (1) the isolate NYN1 (109°57′E and 39°19′N), (2) the isolate A2 (116°10′E and 40°08′N), (3) the isolate T6 (129°57′E and N46°42′N), (4) the isolate X5 (129°46′E and 46°54′N), and (5) the isolate E2 (10°59′E and 47°25′N). They were collected using the wet-sieving method [27]. One gram of soil from each of these sites was added into a pot planting white clover (Trifolium repens L.) with sterilized sands around 3 months. Afterwards, the spores in each pot were collected with wet-sieving, 5 spores according to same morphology of F. mosseae were collected and propagated with white clover again in sands until enough spores were produced. The species and their genetic relationships were identified on the basis of both morphology and the gene SSU (covers the small subunit SSU) rDNA regions [28]; and the Pi transporter gene (PT1) which was identified as phosphate transporter gene, a gene marker for the identification and discrimination of AM fungi in the Genus Glomus [29].” Please see :[29] Feng, J., Huang, Z., Zhang,Y., Rui,W., Lei, X., Li, Z. Beneficial effects of the five isolates of Funneliformis mosseae on the tomato plants were not related to their evolutionary distances of SSU rDNA or PT1 sequences in the nutrition solution production. Plants. 2021. 10,1948. doi: 10.3390/PLANTS10091948
Add to lengend of Fig 1, line 334.
Nutrient solution as listed at Line 236-239
In Fig. 2, the authors talk about changes in root biomass and root length. Only in strain A2 there seems to be a significant increase in root biomass resulting in an increase in root volume and root surface. The authors could incorporate a photo where “root architecture” would be shown, where also root branching could be seen. In Fig.3, when the expression of the specific genes is presented the Y axis refers to “relative expression” in comparison to the control. I believe the authors should clarify in this point if this specific metric has taken into consideration the increase in the root biomass already reported. This would give an estimation of the actual increase in gene expression and not an increase which is due to the increased biomass. To conclude, I believe that in order for this work to be accepted for publication, these specific points need to be clarified as they are really important for the validity of the results presented in Fig. 3. “
We added information in fig 2, line 338-339. The A2, T6, NYN1 shown significant different for root biomass.
From Fig.3:
That is our finding that both root biomass and artichticture as well as gene expressing all were induced by AMFs, with an isolation-specific mannor.
Reviewer 2 Report
The authors did not correct any of my comments.
I think the paper needs some corrections:
- add clear aim of the study to the abstract and introduction,
- add the numbers to all references in the text,
- remove “empty” places from the paper,
- format References section according to Instructions for Authors,
- format all sections of the manuscript according to Instructions for Authors.
You must check your paper very exactly and correct all other mistakes and complete lacking data.
Author Response
Thank you very much for your comments. we corrected the manuscript improved.
Best regards
Zhifang LI
I think the paper needs some corrections:
- add clear aim of the study to the abstract and introduction,
Please see Line 25-27ï¼›line 142-147;
- add the numbers to all references in the text,
- remove “empty” places from the paper,
- format References section according to Instructions for Authors,
We checked.
- format all sections of the manuscript according to Instructions for Authors.
You must check your paper very exactly and correct all other mistakes and complete lacking data.
Yes. We checked and corrected and thank you.
Reviewer 3 Report
Please find the comments and suggestion on the manuscript entitled "Overlapping root architecture and gene expression of nitrogen transporters for nitrogen acquisition of tomato plants colonized with isolates of Funneliformis mosseae in hydroponic production.
- Authors are reporting that the AMF has been isolated, but as per the citation it seems it has been previously isolated and used in previous studies too- Please change the sentence accordingly that it is being previously isolated and used in the studies—otherwise it will be confusing, how it was identified when isolated ----- please change
- The MS is not prepared accordingly as there is no indication of section and subsections numberings--- please include 1, 2, 1.2 etc accordingly------
- Line no- 146- 164- it is not clear in this paragraph what was the volume of growth medium, how it was done , and in what container it was done that 30 gm of AMF was inoculated, --- then it was described in next subheading please sequentially describe the steps as its confusing-- please rewrite---
- Please also include on what medium the AMF was propagated and how it was quantified for the spore count---- please elaborate each steps ---- or if quantified before it must be said so-- Please rewrite this section more clearly----
- Line no- 166- average density of 2.4 plants per m2. Its not good to write 2.4 plants I think it can be either 2 or 3 plants how could it be 2.4 plants please recheck----
- Please clarify about the experimental design, it's not understandable in the current form—
- Please provide some figures to support your results with mycorrhiza colonization and also differences in root architecture ---- with different treatments -----
Author Response
thank you very much for your comments. we discussed and corrected our manuscript. please see below:
Best regards.
Zhifang LI
Comments and Suggestions for Authors
Please find the comments and suggestion on the manuscript entitled "Overlapping root architecture and gene expression of nitrogen transporters for nitrogen acquisition of tomato plants colonized with isolates of Funneliformis mosseae in hydroponic production.
- Authors are reporting that the AMF has been isolated, but as per the citation it seems it has been previously isolated and used in previous studies too- Please change the sentence accordingly that it is being previously isolated and used in the studies—
otherwise it will be confusing, how it was identified when isolated ----- please change corrected see Line 158-162
- The MS is not prepared accordingly as there is no indication of section and subsections numberings--- please include 1, 2, 1.2 etc accordingly------
Added in materials and methods section as it is necessary.
- Line no- 146- 164- it is not clear in this paragraph what was the volume of growth medium, how it was done , and in what container it was done that 30 gm of AMF was inoculated, --- then it was described in next subheading please sequentially describe the steps as its confusing-- please rewrite---
corrected see Line 160-163.
- Please also include on what medium the AMF was propagated and how it was quantified for the spore count---- please elaborate each steps ---- or if quantified before it must be said so-- Please rewrite this section more clearly----
corrected see Line 166.
- Line no- 166- average density of 2.4 plants per m2. Its not good to write 2.4 plants I think it can be either 2 or 3 plants how could it be 2.4 plants please recheck----
Average value, 10m2 had 24 plants.
- Please clarify about the experimental design, it's not understandable in the current form—
We make it more clearly in the section of materials and methods.
- Please provide some figures to support your results with mycorrhiza colonization and also differences in root architecture ---- with different treatments -----
Please see fig 1 and fig 2.
Round 2
Reviewer 1 Report
Third Review on
“Overlapping root architecture and gene expression of nitrogen
transporters for nitrogen acquisition of tomato plants colonized
with isolates of Funneliformis mosseae in hydroponic production”
“This is an original article providing new information on the possible use of various fungal strains of Funneliformis mosseae in ameliorating nitrogen uptake by plants. Although the work is characterized by a certain novelty and provides important information that could be used as a starting point for providing a solution to improving tomato crop yields, there are certain points that need to be addressed and clarified in order to have it published.
The authors have answered to some of my comments of the previous reports. Still, the paper needs to be thoroughly checked by a native speaker, because some sentences do not make any sense). I am including the manuscript highlighted by yellow for the authors to check all the phrases that do not make sense in the manuscript (they are plenty).
In Fig. 2, the authors talk about changes in root biomass and root length. Only in strain A2 there seems to be a significant increase in root biomass resulting in an increase in root volume and root surface. The authors could incorporate a photo where “root architecture” would be shown, where also root branching could be seen.
In Fig.3, when the expression of the specific genes is presented the Y axis refers to “relative expression” in comparison to the control. I believe the authors should clarify in this point if this specific metric has taken into consideration the increase in the root biomass already reported.
To conclude, I believe that, although there has been an improvement in the writing and presentation of the work, further modifications should be made in order for this work to be accepted for publication. I am attaching some points to be addressed by the authors.

Author Response
Dear Sir or Madam,
Thank you very much for your comments and questions. In the case there is any question, please do not hesitate to contact with me.
Best regards!
Zhifang

Reviewer 3 Report
The comments and suggestions on the manuscript entitled "Overlapping root architecture and gene expression of nitrogen transporters for nitrogen acquisition of tomato plants colonized with isolates of Funneliformis mosseae in hydroponic production" has been well defended and addressed by the authors. The changes provided by the authors are sufficient and further I don't have any suggestions or comments.
Thanks for the revised version.
Author Response
Thank you very much for your comments and suggestions.
Best regards.
Zhifang
This manuscript is a resubmission of an earlier submission. The following is a list of the peer review reports and author responses from that submission.
Round 1
Reviewer 1 Report
The comments have been taken into consideration and necessary changes have been made.
Reviewer 2 Report
Review on
“Overlapping root architecture and gene expression of nitrogen
transporters for nitrogen acquisition of tomato plants colonized
with isolates of Funneliformis mosseae in hydroponic production”
This is an original article providing new information on the possible use of various fungal strains of Funneliformis mosseae in ameliorating nitrogen uptake by plants. Although the work is characterized by a certain novelty and provides important information that could be used as a starting point for providing a solution to improving tomato crop yields, there are certain points that need to be addressed and clarified in order to have it published.
My major concern is that the paper is very poorly written and the results are poorly presented. I am including the manuscript highlighted by yellow for the authors to check all the phrases that do not make sense in the manuscript (they are plenty).
There are serious flaws in the way the results are presented and interpreted as well as in the terminology used by the authors. The strains used in this study should be presented at the first paragraph of the Results and a detailed analysis of their isolation procedure as well as the methodology used for their identification should be included. In this way, it would be easier for the reader to understand Fig. 1 which should be more analytically presented and described. The authors need to clarify what exactly does this “nutrient solution” contain as far as the microorganism is concerned. When the filter was used, how is it certain that only spores of this strain are retained and not of other microorganisms too?
In Fig. 2, the authors talk about changes in root biomass and root length. Only in strain A2 there seems to be a significant increase in root biomass resulting in an increase in root volume and root surface. The authors could incorporate a photo where “root architecture” would be shown, where also root branching could be seen. In Fig.3, when the expression of the specific genes is presented the Y axis refers to “relative expression” in comparison to the control. I believe the authors should clarify in this point if this specific metric has taken into consideration the increase in the root biomass already reported. This would give an estimation of the actual increase in gene expression and not an increase which is due to the increased biomass.
To conclude, I believe that in order for this work to be accepted for publication, these specific points need to be clarified as they are really important for the validity of the results presented in Fig. 3.
Reviewer 3 Report
The subject of the manuscript is interesting and consistent with the scope of the Journal.
The abstract faithfully conveys the scope of investigations and conclusions drawn. The keywords correspond well to the scope of the research.
I think the paper needs some corrections:
- add clear aim of the study to the abstract and introduction,
- add the numbers to all references in the text,
- remove “empty” places from the paper,
- format References section according to Instructions for Authors,
- format all sections of the manuscript according to Instructions for Authors.
You must check your paper very exactly and correct all mistakes and complete lacking data of papers.
The manuscript entitled “Overlapping root architecture and gene expression of nitrogen transporters for nitrogen acquisition of tomato plants colonized with strains of Funneliformis mosseae in hydroponic production”, amended according to the revision, can be accepted for publishing in Plants.